# Dietary Regulation of Lipid Metabolism in Gestational Diabetes Mellitus: Implications for Fetal Macrosomia

**DOI:** 10.3390/ijms252011248

**Published:** 2024-10-19

**Authors:** Natalia Frankevich, Vitaliy Chagovets, Alisa Tokareva, Natalia Starodubtseva, Elizaveta Limonova, Gennady Sukhikh, Vladimir Frankevich

**Affiliations:** 1V.I. Kulakov National Medical Research Center for Obstetrics, Gynecology and Perinatology, Ministry of Healthcare of Russian, 117997 Moscow, Russia; v_chagovets@oparina4.ru (V.C.); a_tokareva@oparina4.ru (A.T.); n_starodubtseva@oparina4.ru (N.S.); e_limonova@oparina4.ru (E.L.); g_sukhikh@oparina4.ru (G.S.); v_frankevich@oparina4.ru (V.F.); 2Moscow Center for Advanced Studies, 123592 Moscow, Russia; 3Department of Obstetrics, Gynecology, Perinatology and Reproductology, Institute of Professional Education, Federal State Autonomous Educational Institution of Higher Education I.M. Sechenov First Moscow State Medical University of the Ministry of Health of the Russian Federation (Sechenov University), 119991 Moscow, Russia; 4Laboratory of Translational Medicine, Siberian State Medical University, 634050 Tomsk, Russia

**Keywords:** macrosomia, gestational diabetes mellitus, obesity, lipidome, mass spectrometry, diet therapy

## Abstract

The primary therapeutic approach for managing hyperglycemia today is diet therapy. Lipids are not only a source of nutrients but also play a role in initiating adipocyte differentiation in the fetus, which may explain the development of fetal macrosomia and future metabolic disorders in children born to mothers with gestational diabetes mellitus (GDM). Alterations in the maternal blood lipid profile, influenced by adherence to a healthy diet in mothers with GDM and the occurrence of fetal macrosomia, represent a complex and not fully understood process. The aim of this study was to examine the characteristics of the blood plasma lipid profile in pregnant women with GDM across all trimesters based on adherence to diet therapy. The clinical part of the study followed a case-control design, including 110 women: 80 in the control group, 20 in a GDM group adhering to the diet, and 10 in a GDM group not adhering to the diet. The laboratory part was conducted as a longitudinal dynamic study, with venous blood samples collected at three time points: 11–13, 24–26, and 30–32 weeks of pregnancy. A significant impact of diet therapy on the composition of blood lipids throughout pregnancy was demonstrated, starting as early as the first trimester. ROC analysis indicated high effectiveness of the models developed, with an AUC of 0.98 for the 30- to 32-week model and sensitivity and specificity values of 1 and 0.9, respectively. An association was found between dietary habits, maternal blood lipid composition at 32 weeks, and newborn weight. The changes in lipid profiles during macrosomia development and under diet therapy were found to be diametrically opposed, confirming at the molecular level that diet therapy can normalize not only carbohydrate metabolism but also lipid metabolism in both the mother and fetus. Based on the data obtained, it is suggested that after further validation, the developed models could be used to improve the prognosis of macrosomia by analyzing blood plasma lipid profiles at various stages of pregnancy.

## 1. Introduction

The rise in the incidence of gestational diabetes mellitus (GDM) and its related complications necessitates the search for potentially modifiable risk factors and new solutions.

The prevalence of hyperglycemia in pregnant women in 2019 was about 15.8%, with 83.6% of cases associated with GDM [1]. The risk of developing GDM in the presence of pre-pregnancy obesity increases to 17% [2]. According to WHO estimates, by 2025, the prevalence of obesity worldwide will exceed 21% among women, with a third of all obesity cases occurring in countries such as the USA, China, Brazil, India, and Russia [3,4]. According to the Russian Association of Endocrinologists [5], the frequency of GDM in Russia averages 7%, reaching 16% with concomitant maternal obesity.

Pre-pregnancy BMI and weight gain during pregnancy significantly affect newborn weight. A study by McBain R.D. et al. (2016) found that an increase in BMI by more than 4 kg/m^2^ before a second pregnancy raised the risk of fetal macrosomia (aRR 4.06; 95% CI 2.25–7.34) and GDM (aRR 1.97; 95% CI 1.22–3.19) [6]. It is known that fetal macrosomia complicates 20% of pregnancies in obese women. After the 20th week of pregnancy, maternal hyperglycemia causes hypertrophy of the fetal pancreatic β-cells, leading to fetal hyperinsulinemia and hyperlipidemia. By the time of delivery, this results in a significant increase in fetal fat and the development of macrosomia [7,8,9]. A meta-analysis of 30 studies from 1950 to 2011 showed that the risk of having a large baby doubles when the mother’s BMI is over 30 kg/m^2^ [10]. Obesity not only complicates pregnancy but also increases the risk of GDM. Currently, diet therapy is the main treatment for managing patients with GDM. Although this dietary change recommendation seems simple, it proves to be particularly challenging for many women in real life. Despite their desire to create the best conditions for themselves and the health of their developing fetus, factors such as eating habits, family and cultural dietary customs, hunger, and time constraints become significant obstacles to adhering to prescribed diets [11,12].

Dietary therapy for GDM should adequately meet the nutritional needs of both mother and fetus, aiming to prevent obstetric and perinatal complications. Today, endocrinologists widely recommend a diet excluding carbohydrates with a high glycemic index, easily digestible carbohydrates, and trans fats, with a daily carbohydrate intake of 175 g or at least 40% of the calculated daily caloric intake. This should be under glycemic and urinary ketone monitoring for all pregnant women with GDM [5,13,14,15].

Changes in the profile of circulating lipids in maternal bloodstream, depending on adherence to a rational diet among mothers with GDM and the development of fetal macrosomia, represent a complex and not fully understood process involving both behavioral factors (“eating behavior”) and genetic ones. Lipid metabolism undergoes a number of significant changes during pregnancy. From the early weeks of pregnancy, there is an increase in lipid synthesis and hyperphagia, along with the expansion of maternal fat stores. The adipose tissue shows a significantly heightened receptor response to insulin, leading to the accumulation of circulating lipids. By the end of pregnancy, there is an increase in the lipolytic activity of adipose tissue, and fatty acid synthesis decreases [12,16]. In GDM, there is a disruption of the physiological regulation of these processes, leading to enhanced lipolysis and ketogenesis. Szabo A.J. (2019) proposed that lipids are not only a source of nutrients but also act as triggers for fat cell differentiation in the fetus. This could explain the development of fetal macrosomia, excess body weight, and future obesity in children born to mothers with metabolic disorders [17]. The hypothesis suggests that fatty acids transported across the placenta initiate the transformation of mesenchymal stem cells into fat cells by activating specific transcription factors in the fetus. Accelerated transfer of fatty acids leads to an overproduction of fat cells in the fetus, resulting in excessive fetal weight gain. These fatty acids come from free fatty acids, products of triglyceride breakdown, and polyunsaturated fatty acids from the mother’s phospholipids. Glucose also plays a role as a precursor for alpha-glycerophosphate, which helps esterify most fatty acids [18,19].

The aim of this study was to analyze the lipid profile in the blood plasma of pregnant women with GDM throughout all trimesters, focusing on the impact of dietary therapy.

## 2. Results

### 2.1. Clinical Characteristics

In the first stage, we conducted a comparative analysis of the clinical characteristics of the study groups. The results are shown in Appendix A. No significant differences were found between the main and control groups in terms of age or patient anthropometric data before pregnancy. However, within the group of women with GDM, differences were noted based on whether they followed the prescribed diet. Pregnant women with GDM who did not follow the diet were older (37 years, *p* = 0.08) and had a significantly higher pre-pregnancy weight (81 kg, *p* = 0.004).

Pre-pregnancy BMI and weight gain by delivery were significantly different between the main and control groups (BMI: 22.6 vs. 21.2, *p* = 0.03; weight gain: 14 kg vs. 11 kg, *p* = 0.009). These differences were largely due to the subgroup of women with GDM who did not follow the diet, as they had the highest BMI (26.4, *p* = 0.003) and weight gain (13 kg, *p* = 0.006) among all patients. Notably, the fetal mass at 32 weeks of pregnancy, based on ultrasound, was significantly higher in women who did not adhere to the diet (2132 g vs. 1899 g, *p* = 0.04).

Patients in the main group were more likely to have a cesarean section compared to the control group (60% vs. 34%, *p* = 0.02). Among these, planned cesarean sections were more common (43% vs. 18%, *p* = 0.01), with the highest rate (60%) seen in women with GDM who did not follow the diet. This is likely due to a combination of factors, including the expected large fetal size (>4000 g) and the women’s preferences. Newborn weight in this subgroup was also the highest (3810 g, *p* = 0.14).

Hospital stays for mothers were longer in the main group, regardless of diet adherence (5 days vs. 4 days, *p* = 0.02). However, outcomes for newborns, such as early neonatal complications, Apgar scores at 1 and 5 min, and discharge times, did not differ significantly between the main and control groups.

### 2.2. Lipid Profile Analysis

In the study, HPLC-MS analysis of blood samples from patients in both the main and control groups was conducted at three time points: 11–13, 24–26, and 30–32 weeks of pregnancy. Positive ion mass spectra of plasma lipid extracts are shown on Appendix A.

An OPLS analysis of the lipid profile of the blood plasma of patients with GDM was performed over time (11–13, 24–26, and 30–32 weeks of pregnancy), depending on adherence to diet (Figure 1).

Data clustering showed a significant impact of dietary therapy on blood lipid composition, starting as early as the first trimester of pregnancy. ROC analysis confirmed the effectiveness of the developed models. The lowest AUC was 0.91 for the 11- to 13-week model, with sensitivity and specificity of 0.8 and 1, respectively (Figure 2, Table 1). The highest AUC was 0.98 for the 30- to 32-week model, with sensitivity and specificity of 1 and 0.9, respectively (Figure 2, Table 1).

Lipids with the greatest contribution to sample classification in the OPLS-DA models were identified by a VIP value > 1 (Appendix A). Their levels for the groups under consideration are presented in Figure 3. Comparing groups of pregnant women with gestational diabetes who either adhered to or did not adhere to a diet identified four classes of lipids: phosphatidylcholines (PCs), sphingomyelins, lysophosphatidylcholines, and triglycerides. In the first trimester (11–13 weeks), lipids such as phosphatidylcholines and sphingomyelins were detected, and in the second and third trimesters (24–26 and 30–32 weeks), lysophosphatidylcholines and triglycerides were added to these.

The varying levels of individual lipids within classes, which change dynamically from the first to the third trimester, are of particular interest, especially based on adherence to a diet in gestational diabetes (GD). For example, phosphatidylcholine (PC) of 34:2 (across all trimesters) and sphingomyelin (SM) 46:0, 48:1, 48:0, and 48:2 (in the second and third trimesters) increased in women who did not follow the diet. In contrast, PC 38:3 (across all trimesters) and SM 32:7 and 47:3 (in the second trimester) decreased. Additionally, levels of triglycerides (TGs) of 54:0 (in the second and third trimesters) and 52:0 (in the second trimester), along with lysophosphatidylcholine (LPC) of 16:0 (in the second and third trimesters), remained low in women not adhering to the GD diet. These lipid changes were significant for several lipids, shown in Figure 3. Thus, analyzing the plasma lipid composition helped differentiate between GD patients who followed the diet and those who did not throughout all three trimesters of pregnancy.

The characteristics of the lipid profile may be influenced by both maternal and fetal factors. To build on the existing OPLS analysis, we also included a fetal factor—birth weight. This clinical indicator was chosen based on data showing a high rate of macrosomia (up to 50%) among women with gestational diabetes who did not follow a diet. Of particular interest is the identified link between diet, maternal blood lipid composition at 32 weeks of pregnancy, and the newborn’s weight (Figure 4).

Samples from GDM patients who did not follow the diet and gave birth to children with fetal macrosomia shifted to the right side of the graph. This suggests that changes in the lipid profile during the development of macrosomia are the opposite of those seen with diet therapy. Based on the data, it can be assumed that after further validation, these models could be used to improve predictions of macrosomia development by analyzing the plasma lipid profile at different stages of pregnancy.

## 3. Discussion

It is well known that the primary method for controlling hyperglycemia is diet therapy, which excludes easily digestible carbohydrates. In our study, 30 women with gestational diabetes (the main group) were on diet therapy, but 10 of them admitted to not following the diet. An analysis within the group of women with gestational diabetes, based on diet adherence, revealed several differences between the two subgroups. Women who did not follow the diet were older and had significantly higher pre-pregnancy weight and BMI, suggesting that poor eating habits and psychological challenges might have made it difficult for them to follow the doctor’s dietary recommendations. Lu Liu et al. showed that reducing pre-pregnancy BMI by 10–15% for women with obesity and by 5% for those who are overweight is associated with a significant reduction in adverse perinatal complications (gestational diabetes, hypertension, macrosomia) [20]. Women in the main group were more likely to have a cesarean section than those in the control group. The rate of cesarean deliveries was much higher in patients with gestational diabetes, reaching 60% in those who did not follow the diet. Additionally, these women were more likely to give birth to larger babies. Among the 20 patients who followed the diet, only 3 (15%) had babies with fetal macrosomia, whereas 50% of the women who neglected the diet gave birth to larger babies.

When comparing patients with gestational diabetes mellitus (GDM) who adhered to and did not adhere to a diet, four classes of lipids related to phosphatidylcholines (PCs), sphingomyelins, lysophosphatidylcholines, and triglycerides were identified. At 11–13 weeks, phosphatidylcholines and sphingomyelins were detected, and at 24–26 and 30–32 weeks of pregnancy, the list was supplemented with lysophosphatidylcholines and triglycerides. Phosphatidylcholines and sphingomyelins are major components of cell membranes, involved in signal transduction processes, including the activation of insulin-related pathways [21,22]. Changes in the composition of fatty acids and phospholipids in cells can reduce tissue sensitivity to insulin, which is a key factor in the development of GDM. Phosphatidylcholines can interact with other lipid classes, such as triglycerides and free fatty acids, which can also affect metabolism and the energy status of cells involved in insulin secretion. Changes in lipid composition, including types like phosphatidylcholines and sphingomyelins, may be associated with various metabolic disorders, such as insulin resistance and obesity.

Changes in blood lipid composition due to not following a diet in GDM were observed from the first trimester. A key feature of the plasma lipid profile in these women was the varying levels of specific lipids within classes, which continued to change from the first to the third trimester. Throughout the study, we recorded an increase in phosphatidylcholine (PC) 34:2 and a decrease in PC 38:3 among women who did not follow the diet. A similar pattern was seen in the sphingomyelin group, where SM 46:0, 48:1, 48:0, and 48:2 increased in the second and third trimesters, while SM 32:7 and 47:3 decreased. Additionally, the levels of triglycerides (TGs) 54:0 and 52:0 and lysophosphatidylcholine (LPC) 16:0 remained low in women not adhering to the diet.

Anderson et al. conducted a study examining the lipid profiles in three distinct groups: individuals with overt gestational diabetes mellitus (GDM) (*n* = 18), those with hyperglycemia below the GDM threshold (*n* = 45), and healthy controls (*n* = 43). Their research, published in *PLOS One* in 2014, demonstrated strong positive correlations between free fatty acids (FFAs), phosphatidylcholines (PCs), and lysophosphatidylcholines (LPCs) and the risk of developing GDM [23]. These findings suggest that these lipids, especially PCs, may play a role in disrupting glucose regulatory mechanisms before the onset of hyperglycemia.

In a related study, Liangjian and colleagues investigated lipid profiles in a larger cohort of 817 pregnant women [24]. This study, described in *Diabetes Care* in 2016, employed a novel direct infusion mass spectrometry technique to measure lipids in serum samples collected at 28 weeks of gestation. Participants also underwent an oral glucose tolerance test. Of the 13 lipid species identified, 10 showed significant associations with impaired glucose tolerance. The study highlighted five lipids—TG 50:1, TG 48:1, PC 32:1, PC 40:3, and PC 40:4—as being predictive of GDM independently of maternal age and body-mass index (BMI). This finding underscores the potential of these second-trimester lipids to serve as early indicators of GDM, beyond the traditional risk factors of maternal age and weight.

In 2021, Christopher Papandreou et al. published a study focusing on the relationship between changes in circulating metabolites during weight loss in the context of dietary therapy in patients with obesity. The cohort study included 162 participants who achieved a weight loss of ≥8% during an initial 8-week low-calorie diet, followed by a 12-week observation period. Targeted metabolite profiling (123 metabolites) was performed using three different platforms: proton nuclear magnetic resonance, liquid chromatography–mass spectrometry, and gas chromatography–mass spectrometry. Changes in levels of several types of lipids and citric acid were significantly associated with weight loss, total body fat amount, and abdominal fat distribution. The authors found a decrease in concentrations of LPC 14:0, LPC 20:3, PC 32:2, PC 38:3, and SM 32:2 and an increase in citric acid concentration during the dynamic 12-week observation post-weight loss. The authors conclude that there is an existing connection between weight loss and changes in types of lipids and citric acid. These changes likely reflect lipid metabolism in the body and are an important factor in controlling weight gain and the development of obesity [25]. An earlier study from 2007 also showed an association of PC 34:4 with obesity. The authors demonstrated that a higher plasma level of PC 34:4 was associated with an increased risk of obesity in adults. The absence of dependence on hereditary factors was also demonstrated by conducting a study on a group of monozygotic twins [26]. However, it should be considered that factors like progressive pregnancy, fetal growth, and the presence of already diagnosed GDM can significantly contribute to the maternal blood lipid profile and explain conflicting data on the decrease of several metabolites (PC 38:3, SM 32:7, SM 47:3, TG 54:0, and 52:0, LPC 16:0) over time, against the backdrop of diet cessation in our cohort of pregnant women with GDM. A study from 2020 on lipidomic profiling of 114 plasma samples from overweight or obese pregnant women at three points—before pregnancy, at 15 weeks, and at 35 weeks—revealed changes in 17 lipids, mainly PC and SM, and their connection to GDM [27]. Furthermore, published data (2020) tracked a large cohort of women with GDM post-delivery (2 years of observation) to identify metabolic changes and the development of type 2 diabetes. In that longitudinal study, metabolic changes from the initial inclusion point to 2 years of observation were analyzed as a trajectory of type 2 diabetes progression. When creating a predictive model, the authors identified a distinct metabolic signature in the early postpartum period, which predicted the development of type 2 diabetes (AUC 0.883 (95% CI 0.820–0.945, *p* < 0.001)). The most striking finding at the baseline level was the overall increase in amino acids (AAs), as well as diacylglycerophospholipids and a decrease in sphingolipids and acylalkylglycerophospholipids among women with an onset of type 2 diabetes. Dynamic observation of women with type 2 diabetes showed the preservation or increase of AA regulation and a decrease in sphingolipid and acylalkylglycerophospholipid regulation. The authors found a metabolic signature predicting the transition from GDM to type 2 diabetes in the early postpartum period. They concluded that metabolic dysregulation is present several years before the onset of type 2 diabetes and can be detected in the early postpartum period among women with GDM [28]. Similar data were shown in an experimental study by Moritz Liebmann et al. conducted on two mouse models with and without pre-existing glucose tolerance impairment at the time of pregnancy onset. Mice with initial glucose intolerance exhibited pronounced hyperlipidemia during pregnancy with elevated levels of free fatty acids, triglycerides, and an increased atherogenic index, while hepatic sphingomyelin concentrations decreased in the face of increased plasma sphingosine-1-phosphate (S1P) concentrations. These mice showed impairments in hepatic weight regulation and changes in the metabolism of free fatty acids, accompanied by impaired translocation of fatty acid translocase into the hepatocellular plasma membrane [29]. Thus, elevated concentrations of free fatty acids against a decrease in certain sphingolipid fractions may be a predictor of a more severe course of GDM and high risk of the development of type 2 diabetes.

There is a notable increase in fetal mass, as indicated by ultrasound data at 32 weeks of gestation, in mothers with gestational diabetes mellitus (GDM) who do not follow a diet (1899 (1717; 2110) compared to 2132 (1865; 229), *p* = 0.04). By 32 weeks of pregnancy, this difference may become irreversible for the newborn’s weight, as shown by both the ultrasound fetal data and the results of the OPLS analysis of the plasma lipid profile. According to a study by Robert J. D’Arcy et al., accelerated fetal growth begins long before a GDM diagnosis. The authors emphasize the need for early methods to identify pregnancies at high risk of fetal macrosomia due to GDM, as current methods (like measuring glycated hemoglobin levels in the first trimester) are ineffective for this purpose [30]. This underscores the importance of diet therapy for patients with GDM and adhering to proper nutritional principles during pregnancy planning to prevent fetal macrosomia. It can be inferred that the development of fetal macrosomia in women with GDM heavily relies on adherence to diet therapy: the more compliant the patient, the more likely they are to have babies with normal fetal weight (85% compared to 50%). The samples from patients with fetal macrosomia shifted to the right on the graph, closer to those who did not follow the diet. This indicates that changes in the lipid profile during macrosomia development are opposite to those seen with diet therapy. This finding confirms that diet therapy can normalize not only carbohydrate but also lipid metabolism in both the mother and fetus.

A positive aspect of this work is the long observational period, supported by collected biomaterial (three points: 11–13, 24–26, and 30–32 weeks of pregnancy), which covers the entire pregnancy period for patients with GDM (first, second, and third trimesters). This allowed for the description of unique dynamic changes in blood lipids during pregnancy with GDM and identifying their characteristics depending on the patients’ diet.

The limitations of the study include the small number of patients with GDM (30) and the lack of validation of the method on a large representative number of patients. These limitations currently prevent the practical application of the proposed predictive models.

Despite the small number of patients, our data underscore the potential benefits of metabolomics for prediction, early screening, assessment of neonatal risks, and treatment of GDM. It is important to remember that human metabolism is influenced by both internal (epigenetics and genetic mutations) and external (environment, stress, dietary habits) factors. Future research should focus on addressing the aforementioned limitations, which will help bring our findings closer to clinical practice.

## 4. Materials and Methods

### 4.1. Study Desing

To study the clinical and anamnestic characteristics and obstetric and perinatal outcomes of pregnancies complicated by gestational diabetes mellitus (diet therapy), an observational case–control study was conducted at the National Medical Research Center for Obstetrics, Gynecology, and Perinatology, named after academician V.I. Kulakov, of the Ministry of Healthcare of the Russian Federation. Samples were collected throughout 2023.

The study included 110 women who were observed during pregnancy and childbirth. Blood samples were taken from the cubital vein at 11–13, 24–26, and 30–32 weeks of pregnancy. Informed consent was obtained from all women involved in the study.

Inclusion criteria for the main group (*n* = 30): Caucasian race, singleton pregnancy, newborn weight between 2501 g and 4999 g, presence of gestational diabetes, and patient consent to participate in the study.

Inclusion criteria for the control group (*n* = 80): Absence of gestational diabetes, singleton pregnancy, newborn weight between 2501 g and 3999 g, and patient consent to participate in the study.

Exclusion criteria: Type 1 and 2 diabetes mellitus, any somatic pathology in the decompensation stage, oncological diseases, autoimmune diseases, asthma under medicinal treatment, and multiple pregnancies. The women who requested a switch to insulin therapy were excluded from the study.

A semi-quantitative assessment of plasma lipid levels was conducted using mass spectrometry. For a more detailed analysis, the main group was divided into two subgroups based on adherence to the diet prescribed by an endocrinologist: 20 women followed the diet, while 10 did not. The frequencies of excessive, insufficient, and recommended total weight gain during pregnancy were calculated for women in both groups and subgroups using the criteria established by the Institute of Medicine of the USA (2009).

Several stages were provided for in the study to address the tasks set. The second stage included conducting a longitudinal study (study of time points) among the group of women with gestational diabetes, where blood was collected at three points (at 11–13, 24–26, and 30–32 weeks of pregnancy) and lipidomic analysis of plasma was performed using mass spectrometry (MS), along with statistical analysis of the data obtained depending on the adherence of women with gestational diabetes to the diet prescribed by their doctor.

Ultrasound examinations were performed on all pregnant women at scheduled times (11–14 weeks, 18–21 weeks, and 30–32 weeks of pregnancy).

The diagnosis of GDM is established when fasting venous plasma glucose is detected to be ≥5.1 mmol/L (but <7.0 mmol/L) at any stage of pregnancy, including after an oral glucose tolerance test (OGTT) indicating normal carbohydrate metabolism [5,31,32]. A GDM diagnosis can be based on a single glycemia measurement. This diagnostic criterion applies to the entire gestation period. An OGTT with 75 g of glucose was performed between the 24th and 28th weeks for all pregnant women without pregestational diabetes who did not show carbohydrate metabolism disorders in the first half of the pregnancy or who were not tested for GDM in early pregnancy. During the OGTT, venous plasma glucose was measured: fasting and one and two hours after glucose intake.

### 4.2. Diet Therapy

All pregnant women with gestational diabetes were provided with lifestyle modification recommendations to prevent obstetric and perinatal complications, which included changes in diet and physical activity. The level of evidence for these recommendations is rated as B (evidence reliability level 2).

All pregnant women with gestational diabetes were prescribed a diet therapy excluding high glycemic index (GI) carbohydrates, easily digestible carbohydrates, and trans fats, with a daily carbohydrate intake of 175 g or at least 40% of the calculated daily caloric intake. Glycemic and urinary ketone levels were monitored to adequately meet the needs of the mother and fetus and to prevent obstetric and perinatal complications [5,14]. According to this diet therapy, carbohydrate-containing foods are distributed throughout the day into 3 main meals and 2–3 additional snacks. Each meal should include slowly digestible carbohydrates, protein, mono- and polyunsaturated fats, and dietary fibers. Pregnant women with obesity are advised to limit saturated fats to 10% of their daily fat intake. Caloric intake reduction is recommended for pregnant women with pre-pregnancy obesity based on BMI and those with excessive weight gain during pregnancy, but it should not be less than 1800 kcal daily to prevent ketonuria.

Carbohydrate distribution throughout the day: breakfast 15–30 g, second breakfast 15–30 g, lunch 30–60 g, afternoon snack 15–45 g, dinner 30–60 g, and evening snack 10–15 g, averaging 150–175 g of carbohydrates per day. The interval between meals should be 2.5 to 3 h, with no more than 10 h between the last meal of one day and the first meal of the next. Easily digestible carbohydrates and high GI carbohydrates are completely excluded. To prevent ketonuria or ketonemia, an additional carbohydrate intake (≈12–15 g) is recommended before bedtime or at night [13].

For persistent post-breakfast hyperglycemia, a protein–fat breakfast is recommended with minimal or no complex (or hard-to-digest) carbohydrates. Approved sugar substitutes during pregnancy, such as sucralose and stevia, may be used.

Dosed aerobic physical activities of at least 150 min per week are recommended for pregnant women with gestational diabetes to improve glycemic indicators: daily walking for 10–15 min after meals and before bedtime, provided there are no contraindications.

### 4.3. MS Analysis of Plasma Lipid Extracts

Lipid extracts were obtained using a modified Folch method. A chloroform–methanol mixture (480 μL, 2:1, *v*/*v*) was added to 40 μL blood plasma. The mixture was incubated for 10 min, filtered using filter paper, and 150 μL of an aqueous NaCl solution (1 mol/L) was added to the resulting solution. The mixture was centrifuged at 3000 rpm for 5 min at room temperature. The organic bottom layer containing lipids was collected and dried under a nitrogen stream, then re-dissolved in an acetonitrile-2-propanol mixture (1:1, *v*/*v*) for subsequent mass spectrometric analysis.

The molecular composition of the blood plasma samples was determined using flow injection analysis (FIA) electrospray ionization mass spectrometry with a Maxis Impact qTOF mass spectrometer (Bruker Daltonics, Bremen, Germany) [33,34]. Mass spectra were obtained in positive ion mode within the m/z range of 400–1000 with the following settings: capillary voltage of 4.5 kV, nebulizer gas pressure of 0.7 bar, drying gas flow rate of 6 L/min, and drying gas temperature of 200 °C. A constant flow of eluents A/B at a 1/1 ratio was supplied at a rate of 20 μL/min by a Dionex UltiMate 3000 binary pump, and 5 μL of sample was injected by a Dionex UltiMate 3000 autosampler (ThermoScientific, Bremen, Germany). Acetonitrile–water (60:40, *v*/*v*) with 0.1% formic acid and 10 mM ammonium formate was used as eluent A, and acetonitrile–2-propanol–water (90:8:2, *v*/*v*/*v*) with 0.1% formic acid and 10 mM ammonium formate as eluent B. Tandem mass spectrometry (MS–MS) was used only for specific ions to refine their identification. Data-dependent analysis was used for ion identification.

After the MS analysis, 200 mass spectra obtained during sample elution were averaged, normalized by total ion current (TIC), and transformed into an abundance–m/z table for further processing.

### 4.4. Statistical Analysis

Statistical data processing was performed in RStudio (1.383 GNU) using custom scripts in the R language (4.1.1). Median values (Me) and quartiles (Q1, Q3) were used to describe the quantitative data. Qualitative data are presented as absolute values (%).

Clinical data from three groups—control group, group with GDM and diet therapy, group with GDM and without diet therapy—were compared pairwise. Comparison of qualitative data was performed using χ^2^ tests and comparison of quantitative data was carried out using the Mann–Whitney test for pairwise comparison of groups. The significance threshold in all cases was determined to be 0.05.

Lipid plasma profile from patients with GDM and diet therapy and patients with GDM and without diet therapy from all three points were analyzed using multivariate OPLS-DA (orthogonal projections to latent structures discriminant analysis), which allows for building statistical models using multivariate data to differentiate samples [35]. The influence of individual variables (lipids) on the created model was calculated using the variable influence on projection (VIP) parameter. The lipids with VIP > 1 were labeled as the most significant lipids. Lipids were identified by exact mass using the Lipid Maps database and characteristic tandem mass spectra [36,37].

The quality of OPLS-DA models was calculated by receiver operating curve (ROC) analysis during cross-validation, and ROCs were constructed with the calculation of the area under the curve (AUC).

## 5. Conclusions

The data obtained enhance our understanding of the pathogenetic mechanisms involved in the development of gestational diabetes mellitus (GDM) and fetal macrosomia. Proper nutrition during pregnancy seems to play a crucial role in managing not only carbohydrate metabolism but also lipid metabolism for both the mother and the fetus. Early identification in the first trimester of patients who do not adhere to an appropriate nutritional regimen—coupled with consultations with dietitians and clinical psychologists—can effectively motivate these women to adjust their diets before irreversible changes in fetal weight occur. This proactive approach can help reduce the incidence of large-for-gestational-age newborns and those with macrosomia, decrease the frequency of cesarean sections, improve long-term health outcomes for women with GDM and their children, and promote economic efficiency.

## Figures and Tables

**Figure 1 ijms-25-11248-f001:**
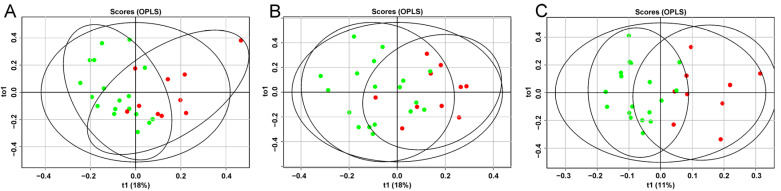
Plots of scores based on the results of OPLS-DA of the lipidomic profile of blood plasma in patients at (**A**) 11–13, (**B**) 24–26, and (**C**) 30–32 weeks of pregnancy who were prescribed dietary therapy. Green dots correspond to patients following the diet, red dots to those not following the diet.

**Figure 2 ijms-25-11248-f002:**
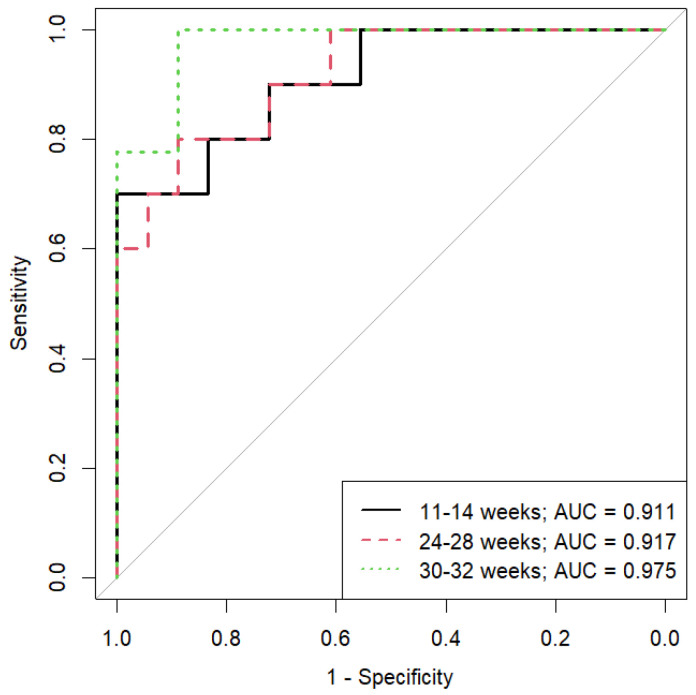
Results of the ROC analysis of OPLS-DA models that differentiate plasma samples from patients adhering to or not adhering to a diet. Samples were taken at 11–13, 24–26, and 30–32 weeks of pregnancy.

**Figure 3 ijms-25-11248-f003:**
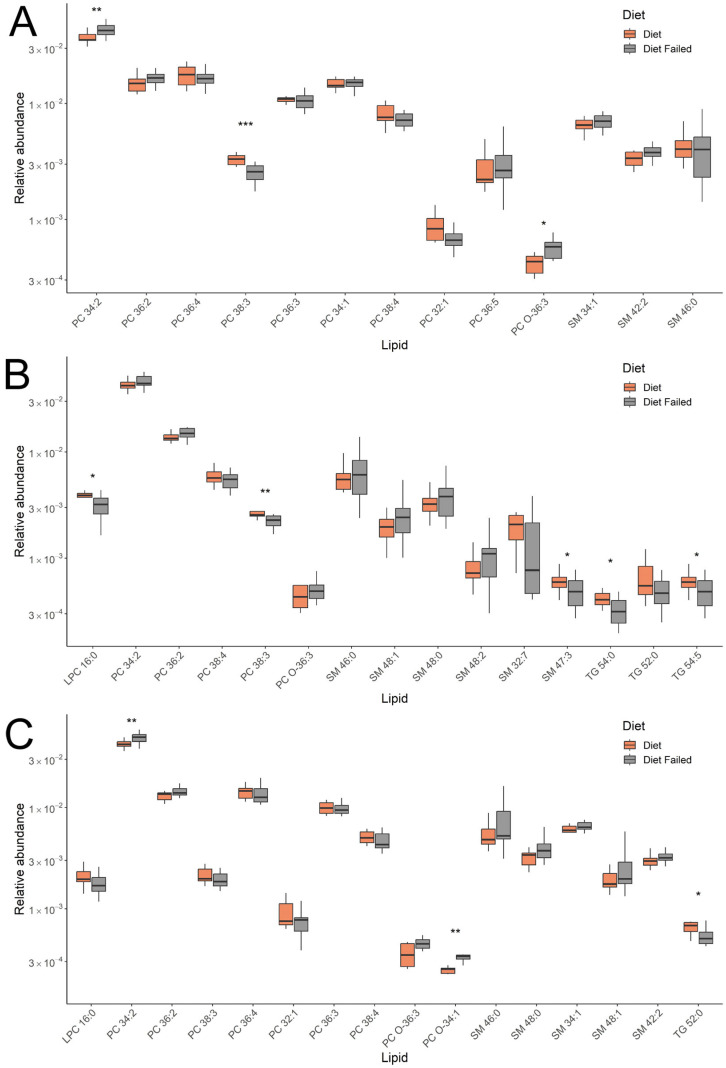
Box plot of lipid levels that were most significant for classifying patients who adhered to the diet and those who did not. The data are provided for the model based on the analysis of samples obtained at (**A**) 11–13 weeks, (**B**) 24–26 weeks, and (**C**) 30–32 weeks of pregnancy. * *p*-value ≤ 0.05; ** *p*-value ≤ 0.01; *** *p*-value ≤ 0.001.

**Figure 4 ijms-25-11248-f004:**
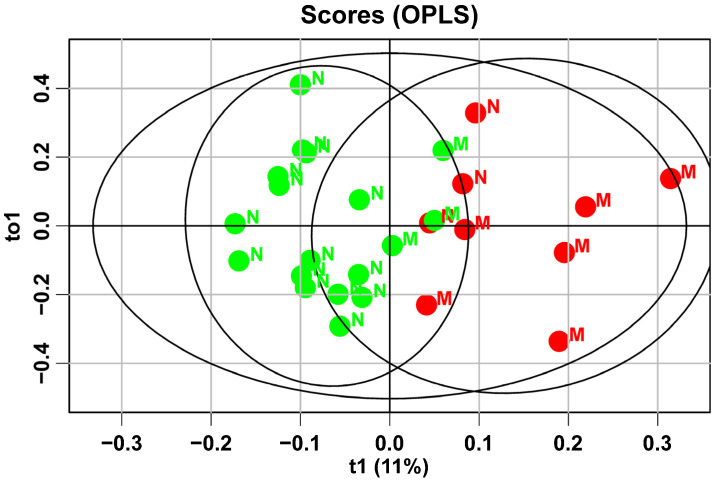
The graph is based on the results of OPLS analysis of the lipidomic profile of plasma from patients at 30–32 weeks of pregnancy who were prescribed diet therapy. Green dots correspond to patients adhering to the diet, and red dots to those not adhering to the diet. N represents patients with normal fetal size, and M represents patients with fetal macrosomia.

**Table 1 ijms-25-11248-t001:** Characteristics of the OPLS-DA models that allow distinguishing blood plasma samples from patients adhering to or not adhering to a diet.

Model	AUC	Threshold	Sensitivity	Specificity
11–14 weeks	0.91	1.54	0.8 (0.6; 1)	1 (0.61; 1)
18–21 weeks	0.92	1.52	0.9 (0.6; 1)	0.94 (0.56; 1)
30–32 weeks	0.98	1.4	1 (0.89; 1)	0.94 (0.78; 1)

## Data Availability

Data are contained within the Appendix A.

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
