# Peer review of "Dietary Regulation of Lipid Metabolism in Gestational Diabetes Mellitus: Implications for Fetal Macrosomia"

_ijms, 2024, doi:10.3390/ijms252011248_

Round 1
Reviewer 1 Report
Comments and Suggestions for Authors
This is an interested study that tries to examine the features of lipid profile of pregnant women with GDM and its adherence to diet therapy.
Introduction: Why this study is important for Russia? What are the rates of obesity and GDM in the population?
Methodology section has some issues:
1. Materials and methods section should be moved after the introduction.
2. The Ethical approval is dated back in 2018. Now we have 2024. How the authors can explain the 6y period in regards to changes in diet and lipidemic profile of the subjects?
3. Include the ethical approval information as well the signed consent approval statement of the patients in this section
3. No information is presented by the authors in regards to the type of diet characteristics and regimen the subjects received.
4. Please provide in the details the lipidemic profile that was examined
5. How were the subjects diagnosed with GDM? Please provide details
6. were the subjects under a medication or insulin for their GDM
7. Did the authors have data to present in regards to the genetic predisposition of these patients?
Results:
1. Please provide data in tables for diet and lips profile of the two groups before and after the diet therapy
2. Move all tables and figures after methodology section
Discussion:
1. move discussion section after results
2. Include a paragraph with the strengths and limitations of the study
Author Response
Reviewer 1.
This is an interested study that tries to examine the features of lipid profile of pregnant women with GDM and its adherence to diet therapy.
Introduction: Why this study is important for Russia? What are the rates of obesity and GDM in the population?
Answer: In the introduction section, the following text was added: The prevalence of hyperglycemia in pregnant women in 2019 was about 15.8%, with 83.6% of cases associated with GDM [1]. The risk of developing GDM in the presence of pre-pregnancy obesity increases to 17% [2]. According to WHO estimates, by 2025, the prevalence of obesity worldwide will exceed 21% among women, with a third of all obesi-ty cases occurring in countries such as the USA, China, Brazil, India, and Russia [3,4]. According to the Russian Association of Endocrinologists [5], the frequency of GDM in Russia averages 7%, reaching 16% with concomitant maternal obesity.
Methodology section has some issues:
- Materials and methods section should be moved after the introduction.
Answer: According to the journal's requirements, the "Materials and Methods" section should come after the Result and Discussion section. We have confirmed this with the Editor, and it is mandatory.
- The Ethical approval is dated back in 2018. Now we have 2024. How the authors can explain the 6y period in regards to changes in diet and lipidemic profile of the subjects?
Answer: The doctor who planned the study in 2018 went on maternity leave, and the collection of samples only began in 2023. However, since the study design did not change, the local ethics committee's approval remained dated 2018. Considering the above, we did not undergo additional approval by the ethics committee for this work.
- Include the ethical approval information as well the signed consent approval statement of the patients in this section
Answer: Information about ethical approval and the patient consent statement are mandatory in the list of documents when submitting an article to the editorial board and were sent by us to the journal's editorial office on the day the article was submitted. The sentence: “ Informed consent was obtained from all women involved in the study” was added to the manuscript.
- No information is presented by the authors in regards to the type of diet characteristics and regimen the subjects received.
Answer: The following text was added to the manuscript (new section Diet Therapy): All pregnant women with gestational diabetes were provided with lifestyle modifica-tion recommendations to prevent obstetric and perinatal complications, which included changes in diet and physical activity. The level of evidence for these recommendations is rated as B (evidence reliability level 2).
All pregnant women with gestational diabetes were prescribed a diet therapy ex-cluding high glycemic index (GI) carbohydrates, easily digestible carbohydrates, and trans fats, with a daily carbohydrate intake of 175 grams or at least 40% of the calculated daily caloric intake. Glycemic and urinary ketone levels were monitored to adequately meet the needs of the mother and fetus and to prevent obstetric and perinatal complications [5,14]. According to this diet therapy, carbohydrate-containing foods are distributed throughout the day into 3 main meals and 2-3 additional snacks. Each meal should include slowly digestible carbohydrates, protein, mono- and polyunsaturated fats, and dietary fibers. Pregnant women with obesity are advised to limit saturated fats to 10% of their daily fat intake. Caloric intake reduction is recommended for pregnant women with pre-pregnancy obesity based on BMI and those with excessive weight gain during pregnancy, but it should not be less than 1800 kcal daily to prevent ketonuria.
Carbohydrate distribution throughout the day: breakfast 15-30 g, second breakfast 15-30 g, lunch 30-60 g, afternoon snack 15-45 g, dinner 30-60 g, and evening snack 10-15 g, averaging 150-175 g of carbohydrates per day. The interval between meals should be 2.5 to 3 hours, with no more than 10 hours between the last meal of one day and the first meal of the next. Easily digestible carbohydrates and high GI carbohydrates are completely ex-cluded. To prevent ketonuria or ketonemia, an additional carbohydrate intake (≈12-15 g) is recommended before bedtime or at night [13].
For persistent post-breakfast hyperglycemia, a protein-fat breakfast was recom-mended with minimal or no complex (or hard-to-digest) carbohydrates. Approved sugar substitutes during pregnancy, such as sucralose and stevia, may be used.
Dosed aerobic physical activities of at least 150 minutes per week were recommended for pregnant women with gestational diabetes to improve glycemic indicators: daily walking for 10-15 minutes after meals and before bedtime, provided there are no contrain-dications.
- Please provide in the details the lipidemic profile that was examined
Answer: The details of the lipid’s profile were added (Supplementary Figure S1).
- How were the subjects diagnosed with GDM? Please provide details
Answer: The following text was added to Material and Methods section:
The diagnosis of GDM is established when fasting venous plasma glucose is detected to be ≥5.1 mmol/L (but <7.0 mmol/L) at any stage of pregnancy, including after an Oral Glucose Tolerance Test (OGTT) indicating normal carbohydrate metabolism [5,31,32]. A GDM diagnosis can be based on a single glycemia measurement. This diagnostic criterion applies to the entire gestation period. An OGTT with 75g of glucose is performed between the 24th and 28th weeks for all pregnant women without pregestational diabetes who did not show carbohydrate metabolism disorders in the first half of the pregnancy or who were not tested for GDM in early pregnancy. During the OGTT, venous plasma glucose was measured: fasting, and one and two hours after glucose intake.
- were the subjects under a medication or insulin for their GDM
Answer: The women who requested a switch to insulin therapy were excluded from the study. The appropriate text was added.
- Did the authors have data to present in regards to the genetic predisposition of these patients?
Answer: The genetic predisposition was not investigated in this study. According to the literature, a woman's risk of developing GDM significantly increases if she has relatives with type 1 or type 2 diabetes. For instance, if type 2 diabetes is diagnosed in a woman's siblings, her risk of GDM increases fourfold. GDM is twice as common in women whose mothers have diabetes. It is often observed in women with genetic mutations that cause Maturity Onset Diabetes of the Young (MODY) [Kleinwechter H, Demandt N. Diabetes in Pregnancy - Type 1/Type 2 Diabetes Mellitus and Gestational Diabetes Mellitus. Dtsch Med Wochenschr. 2016;141(18):1296-1303. https://doi.org/10.1055/s-0042-110555]. In our study, the analysis of patient anamnesis data did not reveal any statistically significant differences between groups. However, among women with GDM who did not adhere to a diet, 50% had close relatives with diabetes, while in the subgroup that followed dietary therapy, 35% had close relatives with type 1 or type 2 diabetes.
Results:
- Please provide data in tables for diet and lips profile of the two groups before and after the diet therapy
Answer: data were provided (Tables S2-S4)
- Move all tables and figures after methodology section
Answer: According to the journal's requirements, the illustrations and tables must be included within the article. We have confirmed this with the Editor, and it is mandatory.
Discussion:
- move discussion section after results
Answer: According to the journal's rules, the distribution of chapters should be as follows: 1. Introduction, 2. Results, 3. Discussion, 4. Materials and Methods, 5. Conclusions. We are following this rule in our article.
- Include a paragraph with the strengths and limitations of the study
Answer: the following paragraph with study limitations was added:
A positive aspect of this work is the long observational period, supported by collected biomaterial (three points: 11-13, 24-26, and 30-32 weeks of pregnancy), which covers the entire pregnancy period for patients with GDM (1st, 2nd, and 3rd trimesters). This allowed for the description of unique dynamic changes in blood lipids during pregnancy with GDM and identifying their characteristics depending on the patients' diet.
The limitations of the study include the small number of patients with GDM (30 people) and the lack of validation of the method on a large representative number of patients. These limitations currently prevent the practical application of the proposed predictive models.
Despite the small number of patients, our data underscore the potential benefits of metabolomics for prediction, early screening, assessment of neonatal risks, and treatment of GDM. It is important to remember that human metabolism is influenced by both internal (epigenetics and genetic mutations) and external (environment, stress, dietary habits) factors. Future research should focus on addressing the aforementioned limitations, which will help bring our findings closer to clinical practice.

Reviewer 2 Report
Comments and Suggestions for Authors
Natalia Frankevich et al., Dietary Regulation of Lipid Metabolism in Gestational Diabetes Mellitus: Implications for Fetal Macrosomia.
-Minor issues: The abbreviations mentioned in the text, which was not expanded what they stand for; figures are of poor in quality, try to improve.
-The authors should include the prevalence of gestational diabetes
-The introduction part of the manuscript should be included “Advantages of Dietary", Dietary benefit of Gestational Diabetes Mellitus (update reference).
-statistical analysis in a dedicated section and the significance of results and all result Statistical analysis should be write more carefully.
-The resolution of the Figure 2 and 4is too low
-Total results are poorly discussed. roughly, the reports in the discussion are not held by the data presented.
-The English should be revised by a native speaker.
Comments on the Quality of English Language-The English should be revised by a native speaker.
Author Response
-Minor issues: The abbreviations mentioned in the text, which was not expanded what they stand for; figures are of poor in quality, try to improve.
Answer: The abbreviations were expanded, figures were improved.
-The authors should include the prevalence of gestational diabetes
Answer: In the introduction section, the following text was added: The prevalence of hyperglycemia in pregnant women in 2019 was about 15.8%, with 83.6% of cases associated with GDM [1]. The risk of developing GDM in the presence of pre-pregnancy obesity increases to 17% [2]. According to WHO estimates, by 2025, the prevalence of obesity worldwide will exceed 21% among women, with a third of all obesi-ty cases occurring in countries such as the USA, China, Brazil, India, and Russia [3,4]. According to the Russian Association of Endocrinologists [5], the frequency of GDM in Russia averages 7%, reaching 16% with concomitant maternal obesity.
-The introduction part of the manuscript should be included “Advantages of Dietary", Dietary benefit of Gestational Diabetes Mellitus (update reference).
Answer: The following text was added to introduction: Currently, diet therapy is the main treatment for managing patients with GDM. Currently, dietary therapy is the primary therapeutic method for managing patients with gestational diabetes (GDM). Although this dietary change recommendation seems simple, it proves to be particularly challenging for many women in real life. Despite their desire to create the best conditions for themselves and the health of their developing fetus, factors such as eating habits, family and cultural dietary customs, hunger, and time constraints become significant obstacles to adhering to prescribed diets [11,12].
Dietary therapy for GDM should adequately meet the nutritional needs of both moth-er and fetus, aiming to prevent obstetric and perinatal complications. Today, endocrinolo-gists widely recommend a diet excluding carbohydrates with a high glycemic index, easi-ly digestible carbohydrates, and trans fats, with a daily carbohydrate intake of 175 grams or at least 40% of the calculated daily caloric intake. This should be under glycemic and urinary ketone monitoring for all pregnant women with GDM [5,13–15]
And the following text was added to introduction:
Changes in the profile of circulating lipids in maternal bloodstream, depending on ad-herence to a rational diet among mothers with GDM and the development of fetal macro-somia, represent a complex and not fully understood process involving both behavioral factors ("eating behavior") and genetic ones. Lipid metabolism undergoes a number of significant changes during pregnancy. From the early weeks of pregnancy, there is an in-crease in lipid synthesis and hyperphagia, along with the expansion of maternal fat stores. The adipose tissue shows a significantly heightened receptor response to insulin, leading to the accumulation of circulating lipids. By the end of pregnancy, there is an in-crease in the lipolytic activity of adipose tissue, and fatty acid synthesis decreases [12,16]. In GDM, there is a disruption of the physiological regulation of these processes, leading to enhanced lipolysis and ketogenesis.
-statistical analysis in a dedicated section and the significance of results and all result Statistical analysis should be write more carefully.
Answer: The Statistical analysis section has been expanded
-The resolution of the Figure 2 and 4is too low
Answer: The quality of Figures was improved
-Total results are poorly discussed. roughly, the reports in the discussion are not held by the data presented.
Answer: The discussion section was improved and extended.
-The English should be revised by a native speaker.
Answer: The manuscript has been revised.

Round 2
Reviewer 1 Report
Comments and Suggestions for Authors
The author have responded positively to all my comments and have made all necessary corrections